# When Does Metadata Conditioning (NOT) Work for Language Model Pre-Training? A Study with Context-Free Grammars

**Rei Higuchi***
The University of Tokyo, RIKEN AIP
higuchi-rei714@g.ecc.u-tokyo.ac.jp

**Ryotaro Kawata***
The University of Tokyo, RIKEN AIP
kawata-ryotaro725@g.ecc.u-tokyo.ac.jp

**Naoki Nishikawa***
The University of Tokyo, RIKEN AIP
nishikawa-naoki259@g.ecc.u-tokyo.ac.jp

**Kazusato Oko**
University of California, Berkeley, RIKEN AIP
oko@berkeley.edu

**Shoichiro Yamaguchi**
Preferred Networks, Inc.
guguchi@preferred.jp

**Sosuke Kobayashi**
Preferred Networks, Inc.
sosk@preferred.jp

**Seiya Tokui**
Preferred Networks, Inc.
tokui@preferred.jp

**Kohei Hayashi**
The University of Tokyo
hayashi.kohei@gmail.com

**Daisuke Okanohara**
Preferred Networks, Inc.
hillbig@preferred.jp

**Taiji Suzuki**
The University of Tokyo, RIKEN AIP
taiji@mist.i.u-tokyo.ac.jp

## Abstract

The ability to acquire latent semantics is one of the key properties that determines the performance of language models. One convenient approach to invoke this ability is to prepend metadata (e.g. URLs, domains, and styles) at the beginning of texts in the pre-training data, making it easier for the model to access latent semantics before observing the entire text. Previous studies have reported that this technique actually improves the performance of trained models in downstream tasks; however, this improvement has been observed only in specific downstream tasks, without consistent enhancement in average next-token prediction loss. To understand this phenomenon, we closely investigate how prepending metadata during pre-training affects model performance by examining its behavior using artificial data. Interestingly, we found that this approach produces both positive and negative effects on the downstream tasks. We demonstrate that the effectiveness of the approach depends on whether latent semantics can be inferred from the downstream task's prompt. Specifically, through investigations using data generated by probabilistic context-free grammars, we show that training with metadata helps improve model's performance when the given context is long enough to infer the latent semantics. In contrast, the technique negatively impacts performance when the context lacks the necessary information to make an accurate posterior inference.

## 1 Introduction

Language Models (LMs) have the adaptability to a wide range of downstream tasks, backed by pre-training on datasets with diverse latent semantics (Gao et al., 2020; Arora & Goyal,

---

*Equal contribution

2023; Hahn & Goyal, 2023; Zhao et al., 2024; Zhang et al., 2025). While data from various sources and domains is one of the keys to LMs' success, it is also a cause of the difficulty in pre-training. Datasets typically contain both structured, factual texts like Wikipedia (Merity et al., 2016), and unstructured, casual texts such as social media posts (Baumgartner et al., 2020). Not only the style but also the topics of the documents vary widely.

To better leverage the diversity of latent semantics, one promising solution is to condition the documents on *metadata*, i.e., high-level information about the source, style, or human preference of the documents that is not directly observable from the documents themselves (Keskar et al., 2019; Korbak et al., 2023; Khalifa et al., 2024). For example, Gao et al. (2025) reported that performance improved on several downstream tasks by prepending the URL or a topic to each web text during LMs' pre-training, even though these cues were not used during inference. This strategy makes the model accessible to the *latent semantics* of data without observing the entire documents, and one might speculate that it invokes the models' ability to better capture these latent semantics, potentially leading to improved performance.

Nevertheless, it remains unclear whether providing metadata serves as a universal solution for LMs' pre-training. Indeed, as reported in Gao et al. (2025) improvements in downstream task performance are not consistent across tasks, and moreover, the addition of metadata does not lead to lower pre-training loss (measured by the cross entropy loss for next-token prediction). Despite the attractiveness and opacity of metadata conditioning, tracing how it influences model performance remains challenging due to the complexity of latent semantics in real-world data. As a result, the community still lacks a comprehensive understanding of when metadata conditioning works.

In this paper, we systematically investigate the impact of prepending metadata during pre-training on the ability of language models. To avoid the difficulty of analysis caused by the complexity of real-world data, we conduct experiments on synthetic data with controllable and well-understood metadata. Specifically, inspired by Allen-Zhu & Li (2023), we generate synthetic data based on context-free grammars (CFGs). We generate data using multiple CFG rules and define metadata as the information that indicates which rule was used to generate each sample. We investigate how the amount of metadata provided affects the model by conducting performance evaluations and probing analyses.

One of our key findings is that metadata conditioning *can negatively affect* performance depending on the task. This is intuitively illustrated in the bottom part of Figure 1.1. As shown in (A1), our experimental results revealed that when the task prompt is short, the model trained with metadata exhibits lower accuracy in inferring latent semantics compared to the model trained without metadata. This results in reduced performance on downstream tasks as in (A2). On the other hand, as shown in (B1), when the task prompt becomes longer, the model trained with metadata becomes capable of inferring latent semantics comparably to the model trained without metadata. In this case, the model trained with metadata demonstrates higher performance on downstream tasks as shown in (B2).

Our contributions can be summarized as follows:

- We investigate the impact of prepending metadata during pre-training. To this end, we generate *a synthetic dataset based on CFGs with controllable metadata*. We conduct pre-training both with and without metadata and compare the models' performance.

- We found *a hidden trade-off in downstream task performance*: by prepending metadata during pre-training, the performance for tasks with long prompts improves, while that for tasks with short prompts does not. We evaluate the trained models using multiple metrics and demonstrate that such a trade-off arises because *metadata conditioning impairs the model's ability to infer latent semantics from limited contexts*.

- To provide an understanding of why models trained with metadata fail to infer latent semantics and how it affects the downstream performance, we introduce a theoretical framework using *predictive distribution with respect to latent semantics*.

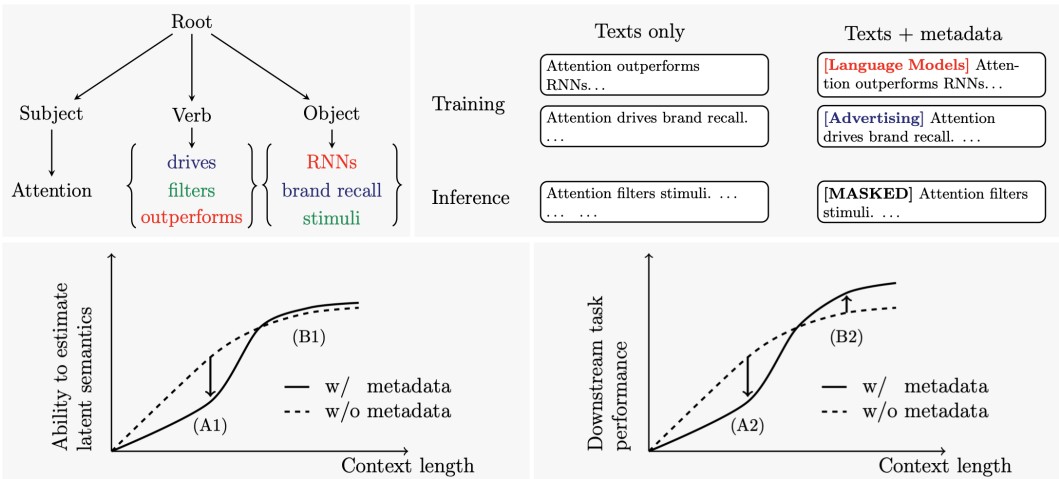

Figure 1.1: Overview and findings. Top: Conceptual illustration of how metadata narrows latent semantics and how it is incorporated into training data. Bottom: Positive and negative effects of using metadata on estimation ability of the latent semantics and downstream task performance.

## 2 Synthetic Dataset with Controllable Metadata

We constructed a dataset using probabilistic context-free grammars (PCFGs) that are systematic and controllable (Allen-Zhu & Li, 2023; Ahuja et al., 2025). We extend the $D$-Level PCFG introduced in Allen-Zhu & Li (2023) by incorporating metadata.

### 2.1 $D$-Level PCFG

First, we explain sentence generation using PCFG. A PCFG is denoted by $\mathcal{G}$, which consists of symbol sets $\{\mathcal{S}_i\}_{i=0}^{D}$ and production rules $\{\mathcal{R}_i\}_{i=0}^{D-1}$. Specifically, we start with a sequence of length one consisting of a unique root symbol $S \in \mathcal{S}_0$, and, for $i = 0, \ldots, D-1$, we sequentially generate a next sequence of symbols of $\mathcal{S}_{i+1}$ from the previous sequence of $\mathcal{S}_i$. Each production rule $\mathcal{R}_i$ has length of 2 or 3, and consists of rules in the form:

$$r = (s_1 \mapsto s_2, s_3, s_4) \quad \text{or} \quad r = (s_1 \mapsto s_2, s_3) \quad \text{for} \quad s_1 \in \mathcal{S}_i \quad \text{and} \quad s_2, s_3, s_4 \in \mathcal{S}_{i+1}. \quad (2.1)$$

At $i$-th step, we replace each symbol in the previous sentence according to the production rule $\mathcal{R}_i$ to generate the next sentence. By repeating this process $D$ times, we obtain the final sequence consisting of symbols in $\mathcal{S}_D$, which we refer to as terminal symbols.

For simplicity, the symbols at the different levels are assumed to be disjoint (i.e., $\mathcal{S}_i \cap \mathcal{S}_j = \varnothing$ for $i \neq j$). The distribution generated with a PCFG $\mathcal{G}$ is denoted by $P(\mathcal{G})$ and its support is denoted by $L(\mathcal{G})$. The procedure for sentence generation is summarized in Algorithm 1.

---

**Algorithm 1** Procedures for sentence generation

1: Set the initial state as $x_0 = S \in \mathcal{S}_0$.
2: **for** $i = 0, \ldots, D-1$ **do**
3:     Keep $x_i = (s_{i,0}, \ldots, s_{i,K_i-1})$ where $s_{i,k} \in \mathcal{S}_i$ for $k = 0, \ldots, K_i - 1$.
4:     **for** $k = 0, \ldots, K_i - 1$ **do**
5:         Randomly sample a rule $r \in \mathcal{R}_i$ whose input is $s_{i,k}$ with uniform probability.
6:         Replace $s_{i,k}$ by the right-hand side of the rule $r$ (e.g., $s_2, s_3, s_4$ or $s_2, s_3$).
7: Output $x_D = (s_{D,0}, \ldots, s_{D,K_D-1})$ where $s_{i,k} \in \mathcal{S}_D$.

---

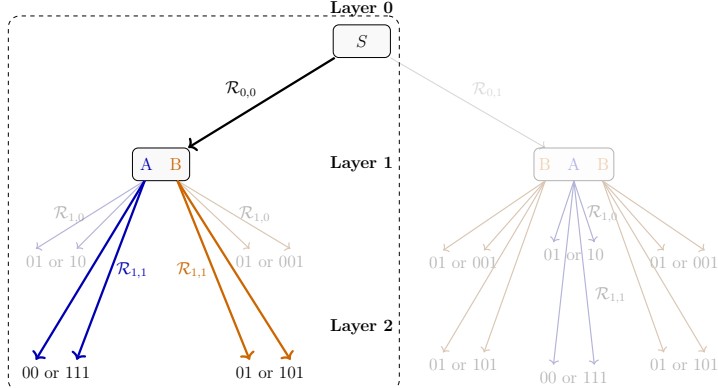

Figure 2.1: An example of 2-Level CFG with hierarchical metadata $(j_0, j_1) = (0, 1)$. Sets of non-terminal symbols are $\mathcal{S}_0 = \{S\}$, $\mathcal{S}_1 = \{A, B\}$ and terminal symbols are $\mathcal{S}_2 = \{0, 1\}$. The grammar is $\mathcal{G}((0, 1)) = (\{\mathcal{S}_i\}_{i=0}^2, \{\mathcal{R}_{0,0}, \mathcal{R}_{1,1}\})$. The support is $L(\mathcal{G}((0, 1))) = \{0001, 00101, 11101, 111101\}$.

## 2.2  $D$-**Level PCFG with hierarchical metadata**

We now introduce the notion of metadata in the context of a $D$-level PCFG, and describe how to generate a sequence under metadata constraints. The overview is shown in Figure 2.1.

As we conceptualize metadata as high-level information underlying the sentence generation process, we operationalize it as a set of meta-level information that modulate the production rules $\{\mathcal{R}_i\}_{i=0}^{D-1}$. Specifically, we suppose that there are multiple choices $(\mathcal{R}_{i,j})$ of each production rule $\mathcal{R}_i$. At each level $i$, we select one of these rules, whose index is represented by $j_i$. We formally define the metadata as the sequence of indices $(j_0, \ldots, j_{D-1})$ corresponding to the selected production rules. Once the rule sequence $\{\mathcal{R}_{i,j_i}\}_i$ is determined, the PCFG grammar is constructed as $\mathcal{G}((j_0, \ldots, j_{D-1})) = (\{\mathcal{S}_i\}_{i=0}^D, \{\mathcal{R}_{i,j_i}\}_{i=0}^{D-1})$, from which we generate a sequence. Note that different sequences use different sample of $(j_0, \ldots, j_{D-1})$.

In this way, within the PCFG framework, we define metadata as meta-information that specifies the choice of production rules. Although metadata directly defines the content of the generated sentence, it constrains the possible sentence structures compared to the full set of rules $\{\cup_{j_i} \mathcal{R}_{i,j_i}\}_{i=0}^{D-1}$. This corresponds to how, in real-world settings, labels such as source, style, or human preference roughly partition the space of plausible sentences.

### 2.3  Generation of data

Finally, we describe how datasets are generated. To investigate how the amount of metadata provided affects the learning of LMs, we manipulate the level-wise availability of metadata labels $j_i$ in PCFG. We refer to providing metadata up to depth $D_M (\leq D)$ as supplying the partial metadata sequence $(j_0, \ldots, j_{D_M-1})$ from the full metadata $(j_0, \ldots, j_{D-1})$. Roughly, $D_M$ quantifies the extent of available metadata, whereas $D - D_M$ represents the range over which the model must navigate the rule space without metadata guidance. In practice, this corresponds to controlling the extent of specificity with which sentences are conditioned.

To provide metadata up to depth $D_M$, we prepend $D_M$ metadata tokens $(j_0, \ldots, j_{D_M-1})$ followed by $D - D_M$ masked tokens to the sentence $x \sim P(\mathcal{G}((j_0, \ldots, j_{D-1}))) = P((\{\mathcal{S}_i\}_{i=0}^D, \{\mathcal{R}_{i,j_i}\}_{i=0}^{D-1}))$. The training dataset is constructed by mixing sentences augmented with metadata as described above and sentences prefixed with $D$ masked tokens with equiprobability. For further details, please refer to Appendix A. The inference dataset, as well as the metadata-less training dataset used for comparison, is constructed by simply prepending $D$ masked tokens to each generated sequence. The purpose of inserting non-informative masked tokens is to maintain a constant sequence length while varying the amount of provided information. We take care to control this aspect, as prior work has shown that even neutral tokens, such as sink tokens, may influence learning (Darcet et al., 2024; Gu et al., 2025).

## 3 Experiments

We now investigate when and how metadata conditioning affects model behavior through experiments on multiple tasks and metrics. We choose our Transformer model by following Allen-Zhu & Li (2023), who similarly trained on CFGs. Specifically, we adopt the LLaMA architecture (Touvron et al., 2023) with 12 layers, 12 attention heads, RoPE (Su et al., 2024), and a hidden size of 768. The data generation process, described in Section 2, uses PCFG with hierarchical level $D = 5$. Each $\mathcal{R}_i$ is selected from one of $\mathcal{R}_{i,0}$ or $\mathcal{R}_{i,1}$ with equiprobability. Only two candidates per level may appear limited, but each metadata label exponentially narrows down the set of possible grammars. Roughly speaking, full metadata reduces the ambiguity in the generative process by a factor of $2^5 = 32$ compared to the metadata-free case. Therefore, this setup is enough to meaningfully examine the influence of metadata.

We compare the results by varying the depth of prepended metadata $D_M \in \{0, 1, 3, 5\}$, where larger $D_M$ means richer conditioning. We pre-train the model using next-token prediction (NTP) with cross-entropy loss. The loss is computed only for terminal symbols and the EOS token, excluding the BOS token and metadata. Therefore, the only difference among training setups is whether metadata is included, and no additional supervision is provided. See Appendix A for further discussion.

### 3.1 Training with or without Metadata *Does Not* Affect Next-Token Prediction Loss

**Test loss *without* metadata.** As a preliminary result, we examine next-token prediction performance. The left of Figure 3.1 shows the test loss across training, evaluated under inference without metadata. Although our dataset is more complex than that of Allen-Zhu & Li (2023) due to hierarchical branching, the models reduce the loss well. Notably, we find no significant difference in the loss between the models trained with metadata ($D_M \geq 1$) and without it ($D_M = 0$), which aligns with the insight from Gao et al. (2025).

**Test loss *with* metadata.** The right of Figure 3.1 shows the loss *when inference is performed on data with prepending metadata*. We can see that it is substantially smaller than the loss in the left figure (inference without metadata). This indicates that models trained with metadata conditioning can indeed perform better during inference, if metadata is available. This observation contrasts with the case where metadata is not used at inference.

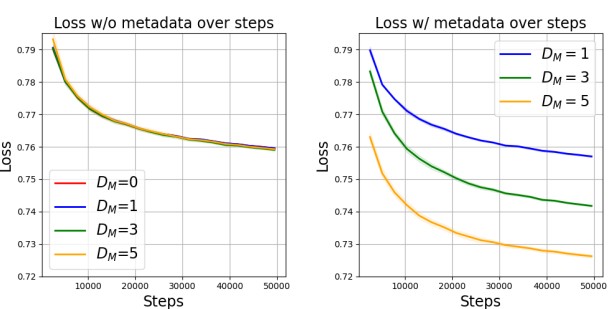

Figure 3.1: Test loss of next-token prediction during training. Left: *without* metadata. Right: *with* metadata.

We summarize these results in the following important observation[1]:

> **Observation.** For average next-token prediction loss, metadata conditioning improves performance *only* when metadata is available at inference time. If metadata is not accessible at inference, no significant improvement is observed.

### 3.2 Training with Metadata *Degrades* the Accuracy of Metadata Prediction

Average next-token prediction loss provides only a coarse measure of model behavior and fails to capture fine-grained differences. This limitation makes it difficult for previous studies to understand the mechanism of metadata conditioning. However, our controlled setup enables a more detailed investigation. To this end, to examine how well the trained model can properly infer the latent semantics from the document, we consider a probing

---

[1]We also examine calibration errors in Appendix C, which further supports this finding.

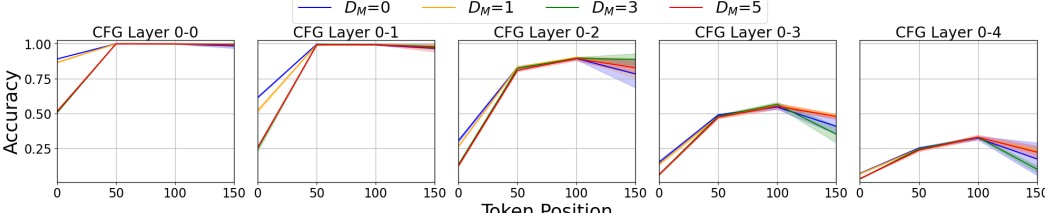

Figure 3.2: Metadata-probing accuracy. We plot the transition with respect to the token position. We used the 12th layer of a Transformer and each column corresponds to the depth of the CFG tree. The $d$-th column ($d = 1, 2, 3, 4, 5$) represents the accuracy of classifying branches of the CFG tree from layer 0 up to layer $d - 1$. The chance rate of correctly predicting the metadata from layer 0 to layer $d - 1$ is $1/2^d$.

task (Conneau et al., 2018; Belinkov, 2022; Allen-Zhu & Li, 2023) that predicts the metadata in order to investigate whether the trained models can properly extract the latent semantics.

**Details of probing.** Let $l \in \{0, \ldots 11\}$ be a layer index of a trained model, and $d \in \{0, \ldots, D-1\}$ be a level of the PCFG rule. Given an input generated from $\mathcal{G}((j_0, \ldots, j_{D-1}))$, let $h_{l,i}$ denote the average of hidden states over 5 consecutive tokens starting from position $i$ at layer $l$. For each $(l, d, i)$, we train $d$ distinct linear binary classifiers (with all model parameters frozen); the $d'$-th classifier predicts the metadata $j_{d'}$ from $h_{l,i}$. We define *metadata-probing accuracy* as the rate at which all $d$ classifiers return the correct labels. The hidden states of later tokens generally aggregate information over longer ranges of the input, and thus tend to yield better probing performance. We therefore evaluate accuracy across various positions $i$ to understand how much context is required to recover the metadata. We also conduct probing across multiple layers (specifically, the 1st, 5th, 8th, and 12th layers) to gain a comprehensive view of how metadata information is distributed throughout the network, while we will present the result only for the 12th layer since the results from other layers exhibit similar results. The results for the other layers can be found in Appendix D.

**Results.** The results are summarized in Figure 3.2. Before seeing the results, one might intuitively expect that a model trained with metadata conditioning would associate latent semantics with document content, thereby improving metadata probing accuracy. In fact, however, *metadata conditioning worsened label-probing accuracy* rather than improving it. In particular, when the prompt length is short, the accuracy of models trained with metadata is notably worse than that of models trained without it. When metadata of depth 3 or 5 is prepended during pre-training, the accuracy at prompt length 5 (token position $i = 0$) is around chance level. This indicates that *models trained with metadata cannot infer latent semantics during inference if the prompt provides too little information.*

In contrast, as the prompt length increases, models trained with metadata become capable of accurately predicting the metadata. Indeed, the large accuracy gap (between $D_M = 0$ and $D_M = 3, 5$) seen at prompt length 5 largely disappears at prompt lengths of 50 or 100. This suggests that models trained with metadata can extract latent semantics from the sequence, as long as the prompt is sufficiently long to provide the necessary information. In summary, we obtain the following finding through the probing:

---

**Finding 1.** *Metadata conditioning leads models to fail to infer latent semantics.* In particular, if the prompt length is short, models trained with metadata conditioning cannot infer latent semantics, while models trained without metadata can. When *the prompt becomes longer*, information in the prompt increases, which allows both types of models to *better capture latent semantics.*

---

In Section 4, we provide a theoretical framework that explains why such counterintuitive performance degradation occurs.

Table 3.1: Grammatical accuracy for various prompt length.

| depth of metadata | length of prompt | | | | |
|---|---|---|---|---|---|
| | 1 | 5 | 10 | 25 | 50 |
| 0 | $0.000 \pm 0.000$ | $0.810 \pm 0.059$ | $0.834 \pm 0.036$ | $0.816 \pm 0.027$ | $0.834 \pm 0.019$ |
| 1 | $0.200 \pm 0.400$ | $0.802 \pm 0.159$ | $0.820 \pm 0.065$ | $0.884 \pm 0.027$ | $0.860 \pm 0.018$ |
| 3 | $0.000 \pm 0.000$ | $0.072 \pm 0.088$ | $0.884 \pm 0.036$ | $0.878 \pm 0.029$ | $0.856 \pm 0.019$ |
| 5 | $0.000 \pm 0.000$ | $0.000 \pm 0.000$ | $0.740 \pm 0.045$ | $0.830 \pm 0.030$ | $0.824 \pm 0.034$ |

## 3.3 Training with Metadata *Degrades* the Downstream Performance for Short Prompts

Finally, we analyze the effect of metadata conditioning on downstream task performance. Although previous studies such as Gao et al. (2025) report improvements in downstream tasks, they do not offer a detailed explanation for this effect. To gain deeper understanding, we perform sequence generation and employ a metric called grammatical accuracy, introduced by Allen-Zhu & Li (2023), which measures how accurately a model generates grammatically correct sentences.

**Details of grammatical accuracy.** Here, we describe how to calculate the metric. First, we randomly provide a model the initial part of a sequence and evaluate whether it can generate a grammatically correct sequence. Specifically, we first obtain $n$ i.i.d. sample sequences $\{(x_{i,1}, \ldots, x_{i,K_i})\}_{i=1}^{n}$ from $L(\mathcal{G})$, where $\mathcal{G} := (\{\mathcal{S}_i\}_{i=0}^{D}, \{\cup_{j_i} \mathcal{R}_{i,j_i}\}_{i=0}^{D-1})$ represents the mixture PCFG. Then, we extract the first $l_p$ tokens $(x_{i,1}, \ldots, x_{i,l_p})$ as prompts and let the model generate token sequences autoregressively until it outputs [EOS]. We denote the generated sequences by $(y_{i,l_p+1}, \ldots, y_{i,M_i})$ and we define $z_i := (x_{i,1}, \ldots, x_{i,l_p}, y_{i,l_p+1}, \ldots, y_{i,M_i})$ for $i = 1, \ldots, n$. Using $z_i$ $(i = 1, \ldots, n)$, we define the grammatical accuracy (GA) as $\text{GA}(\mathcal{G}) = \sum_{i=1}^{n} \frac{\mathbb{1}[z_i \in L(\mathcal{G})]}{n}$. The value $\mathbb{1}[z_i \in L(\mathcal{G})]$, i.e., whether $z_i$ is a correct sequence of grammar $\mathcal{G}$, can be determined by the (deterministic) CYK algorithm. As seen in the previous subsection, the performance of models trained with metadata is highly dependent on prompt length. Therefore, we present this metric across various prompt lengths $l_p$.

**Results.** The results are shown in Table 3.1. Surprisingly, seemingly contrary to the findings of Gao et al. (2025) that showed improved performance on downstream tasks, metadata conditioning led to decreased grammatical accuracy when the prompt length was short. However, as the prompt length increases, models trained with metadata conditioning recover, achieving performance comparable to or better than those trained without metadata.

This pattern is reminiscent of **Finding 1**; models trained with metadata struggle to infer latent semantics from short prompts, resulting in degraded task performance. On the other hand, when prompts contain sufficient information, models successfully infer metadata. As previously noted (**Observation**), metadata-conditioned models achieve lower loss when metadata is explicitly available; thus, metadata inferred from richer prompts similarly enhances downstream task performance. These results lead to the following finding:

> **Finding 2.** *Metadata conditioning degrades the downstream task performance if the task prompt is short*. This is because models trained with metadata cannot infer latent semantics from the short prompt. On the other hand, *for tasks with long prompts, metadata conditioning improves the performance* since the models can easily extract latent semantics from the prompts.

# 4 Interpretation of Experimental Results and Findings

We compare training with and without metadata in how the model implicitly learns latent semantics. Without metadata, the model needs to learn the hierarchical mixture of grammars directly from data, which is challenging due to the diverse structure of possible grammars.

With metadata, grammar inference becomes easier, but accurate prediction of the metadata itself becomes necessary. We highlight this trade-off in inference and learning.

### 4.1 A framework for Latent Semantics Acquisition via Marginalization in the Presence or Absence of Metadata

#### 4.1.1 Training without metadata

We begin by formalizing the role of latent semantics when training is performed without metadata. Given the prompt $x$, the predictive distribution of the output $y$ can be decomposed as

$$p(y|x) = \int \underbrace{p(y|x, \mathcal{G}_\mathrm{L})}_{\text{Likelihood}} \underbrace{p(\mathcal{G}_\mathrm{L}|x)}_{\text{Posterior}} \, \mathrm{d}\mathcal{G}_\mathrm{L},$$

where $\mathcal{G}_\mathrm{L}$ (e.g. PCFG) is latent semantics. For simplicity, we focus on the situation where $\mathcal{G}_\mathrm{L}$ is a PCFG. This decomposition from the principle of marginalization means that the output $y$ can be viewed as being generated through two steps:

(A) Sampling a PCFG (latent semantics) $\mathcal{G}_\mathrm{L}$ from posterior $p(\mathcal{G}_\mathrm{L}|x)$.

(B) Generating $y$ conditioned on both $x$ and $\mathcal{G}_\mathrm{L}$ using the likelihood function $p(y|x, \mathcal{G}_\mathrm{L})$.

When trained without providing the metadata $j$, the model implicitly approximates the likelihood $p(y|x, \mathcal{G}_\mathrm{L})$ and the posterior $p(\mathcal{G}_\mathrm{L}|x)$ internally. Prediction of $p(\mathcal{G}_\mathrm{L}|x)$ is not trivial even if we know the candidates for PCFG $\{\mathcal{G}(j) \mid j \in \{0,1\}^D\}$ because we get samples from different hierarchical CFGs: The total support of the text data is $\bigcup_{j \in \{0,1\}^D} L(\mathcal{G}(j))$.

#### 4.1.2 Training with metadata

On the other hand, when we train the model with metadata $j = (j_0, \ldots, j_{D-1})$, the predictive distribution of the output $y$ in inference can also be decomposed as

$$p(y|x) = \int \underbrace{p(y|x, j)}_{\substack{\text{Likelihood} \\ \text{in training w/ } j}} \underbrace{p(j|x)}_{\text{Posterior of } j} \, \mathrm{d}j = \int \int \underbrace{\underbrace{p(y|x, \mathcal{G}_\mathrm{L})}_{\text{Likelihood}} \underbrace{p(\mathcal{G}_\mathrm{L}|x, j)}_{\text{Posterior given } j}}_{\text{Likelihood in training with } j} \, \mathrm{d}\mathcal{G}_\mathrm{L} \underbrace{p(j|x)}_{\text{Posterior of } j} \, \mathrm{d}j.$$

Note that $p(y|x, \mathcal{G}_\mathrm{L}, j) = p(y|x, \mathcal{G}_\mathrm{L})$ in this setting (we do not need $j$ once we know the grammar $\mathcal{G}_\mathrm{L}$). This decomposition can be expressed in three steps:

(A-1) Sampling metadata $j = (j_0, \ldots, j_{D-1})$ from its posterior $p(j|x)$.

(A-2) Sampling a PCFG $\mathcal{G}_\mathrm{L}$ from *its richer posterior given metadata* $p(\mathcal{G}_\mathrm{L}|x, j)$.

(B) Generating $y$ conditioned on both $x$ and $\mathcal{G}_\mathrm{L}$ using the likelihood function $p(y|x, \mathcal{G}_\mathrm{L})$.

In this case, the model focuses on the tasks (A-2) and (B) because the information of $j$ is provided. Therefore, $p(y|x, j)$ is implicitly constructed in the model. As the training dataset includes both samples with and without metadata, (A-1) and (A-2) are learned separately.

### 4.2 Understanding Trade-offs in Metadata Conditioning through a Formal Framework

The proposed framework highlights both positive and negative effects of metadata conditioning on inference:

✓ The advantage of metadata conditioning is that learning the richer posterior given metadata $p(\mathcal{G}_\mathrm{L}|x, j)$ is facilitated. Once we obtain the candidates for CFG $\{\mathcal{G}(j) \mid j \in \{0,1\}^D\}$, predicting $p(\mathcal{G}_\mathrm{L}|x, j)$ in (A-2) becomes much easier than estimating $p(\mathcal{G}_\mathrm{L}|x)$ in (A).

✗ The drawback of metadata conditioning is that the estimation performance of posterior $p(j|x)$ in (A-1) deteriorates. When inference is performed without metadata, the posterior of the latent semantics $p(\mathcal{G}_L|x) = \int p(\mathcal{G}_L|x,j)p(j|x)\mathrm{d}j$ degenerates, making it more difficult to capture the latent semantics accurately.

These frameworks allow us to interpret the previous experimental results as follows:

**(i) Interpretation of Observation.**  With metadata conditioning, prediction of $p(j|x)$ is less efficient, but prediction of $p(\mathcal{G}_L|x,j)$ is more efficient than without metadata. It implies a trade-off in prediction of $p(j|x)$ and $p(\mathcal{G}_L|x,j)$. Therefore, the average next token prediction loss remains consistent regardless of whether metadata conditioning is used.

**(ii) Interpretation of Finding 1.**  We found that, training with metadata, prediction of $p(j|x)$ becomes degenerated in **Finding 1**. This negatively affects prediction of $p(\mathcal{G}_L|x)$ as discussed in **(i)**. However, when the task prompt $x$ is sufficiently rich, predicting $p(j|x)$ is easy and the negative effect of metadata conditioning on prediction of $p(\mathcal{G}_L|x)$ becomes small.

**(iii) Interpretation of Finding 2.**  When the prompt $x$ is limited, it is difficult to predict $p(j|x)$ as mentioned in **(ii)**. In this case, the no-metadata model predicts $p(\mathcal{G}_L|x)$ more accurately and achieves higher grammatical accuracy (downstream performance). When $x$ is rich, predicting $p(j|x)$ becomes easier. In this situation, there is no difference between the no-metadata and metadata settings in how well $p(j|x)$ is learned. As the metadata-conditioned model efficiently predicts $p(\mathcal{G}_L|x,j)$, it can implicitly construct $p(\mathcal{G}_L|x) = \int p(\mathcal{G}_L|x,j)p(j|x)\mathrm{d}j$ more accurately. Therefore, the with-metadata setting achieves higher grammatical accuracy.

# 5   Related Work

**Control of Generated Text:** While instruction-tuned models can follow user-written prompts in natural language, relying solely on such instructions often leads to ambiguity or inconsistent outputs. As a complementary direction, Keskar et al. (2019), Chan et al. (2020), Dathathri et al. (2019), and Aghajanyan et al. (2021) address this by training models with metadata that encode attributes such as domain, style, task, or source of information. Korbak et al. (2023) incorporate preference scores derived from human judgments as conditioning signals, resulting in reduced toxicity even in unconditional generation. After such training, metadata can be used to more reliably steer the model toward generating text in a specific genre or a writing style.

**Efficiency and Performance Improvement in LLM-training:** LLM training methods that incorporate multiple pre-training and post-training stages with diverse forms of labeling have demonstrated improved downstream performance. For example, Aghajanyan et al. (2021) leverage hyper-textual structures, Gao et al. (2025) integrate metadata such as URLs, and Korbak et al. (2023) utilize human preference scores during pre-training. Zhang et al. (2024) reduce catastrophic forgetting during post-training by conditioning on short topic-related hints. Allen-Zhu & Li (2024) show that training with a special token marking "junk" documents enables the model to allocate its capacity more efficiently. Krasheninnikov et al. (2023) demonstrate that tagging the source reliability helps the model acquire implicit meta-learning capabilities and prioritize high-quality information during fine-tuning.

These studies suggest that metadata can enhance not only controllability but also performance. However, how such conditioning influences model behavior—particularly when metadata is used at pre-training but unavailable at inference—has remained unclear. Our study directly addressed this issue through controlled experiments using PCFGs inspired by Allen-Zhu & Li (2023) and revealed that the benefits critically depend on the relationship between latent semantics and the information provided by prompts at inference time.

# 6 Conclusion

In this study, we systematically investigated the impact of prepending metadata during language model pre-training, utilizing synthetic data generated from context-free grammars with controllable metadata. By comparing models trained with and without metadata, and guided by a theoretical framework, we aimed to elucidate the conditions under which metadata conditioning affects model performance.

Our key finding reveals a trade-off: metadata conditioning enhances performance on downstream tasks with sufficiently long task prompts that allow for the prediction of latent semantics, but it degrades performance on tasks with task prompts too short to predict latent semantics. These findings highlight the need for a more nuanced approach to metadata conditioning in practical pre-training, offering a crucial practical guideline: simply maximizing metadata is not always beneficial. Instead, strategies should be tailored based on the expected nature of downstream tasks. Conditioning with excessive metadata beyond what is potentially inferable from the typical prompt can be counterproductive for downstream tasks.

Future work should focus on developing pre-training strategies that mitigate this trade-off, potentially by adaptively adjusting the use of metadata based on task complexity. Additionally, exploring the impact of different metadata types on model performance will provide valuable insights into optimizing metadata conditioning for diverse applications.

## Acknowledgments

NN was partially supported by JST ACT-X (JPMJAX24CK) and JST BOOST (JPMJBS2418). RK and RH were partially supported by JST CREST (JPMJCR2115). TS was partially supported by JSPS KAKENHI (24K02905) and JST CREST (JPMJCR2015). This research is supported by the National Research Foundation, Singapore, Infocomm Media Development Authority under its Trust Tech Funding Initiative, and the Ministry of Digital Development and Information under the AI Visiting Professorship Programme (award number AIVP-2024-004). Any opinions, findings and conclusions or recommendations expressed in this material are those of the author(s) and do not reflect the views of National Research Foundation, Singapore, Infocomm Media Development Authority, and the Ministry of Digital Development and Information.

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

# A  More Details of the Experimental Setting

**Training Setting.**  The number of data samples used for pre-training is 5,000,000, and the training is conducted over a single epoch. We note that the difference in results between the models with and without metadata is not due to overfitting. For batch size, optimizer, and learning rate, we followed the setup in Allen-Zhu & Li (2023). Specifically, we used a batch size of 96, AdamW optimizer with $\beta = (0.9, 0.98)$, weight decay of 0.1, and a learning rate of 0.0003.

**Data Generated from $D$-Level PCFG.**  Here, we provide a full explanation of the data generated by the procedure described in Section 2.3. The resulting input sequence during training with metadata is given by

$$[\text{BOS}] \quad \tilde{j}_0 \quad \tilde{j}_1 \quad \cdots \quad \tilde{j}_{D-1} \quad s_{D,0} \quad s_{D,1} \quad \cdots \quad s_{D,K_D-1} \quad [\text{EOS}]. \tag{A.1}$$

Here, $\tilde{j}_0, \tilde{j}_1, \ldots, \tilde{j}_{D-1}$ are the tokens that sometimes provide metadata and otherwise become mask tokens. Specifically, they are given as

$$\tilde{j}_0 \quad \tilde{j}_1 \quad \cdots \quad \tilde{j}_{D-1} := \begin{cases} j_0 \quad \cdots \quad j_{D_M-1} \quad \underbrace{[\text{MASK}] \cdots [\text{MASK}]}_{D-D_M \text{ times}} & \text{with probability} = 0.5, \\ \underbrace{[\text{MASK}] \cdots \quad \cdots \quad \cdots \quad \cdots [\text{MASK}]}_{D \text{ times}} & \text{with probability} = 0.5, \end{cases}$$

where $j_i$ ($i = 0, \ldots, D-1$) represents a metadata and $D_M$ represents the depth up to which the metadata is fed into the model.

# B  The loss of next-token prediction for various token positions.

Table B.1: The loss of next-token prediction for various token positions.

| depth of metadata | Position of the tokens | | | | |
|---|---|---|---|---|---|
| | 0 | 5 | 10 | 25 | 50 |
| 0 (no metadata) | 0.9224 | 0.8867 | 0.7244 | 0.7471 | 0.7483 |
| 1 | 0.9224 | 0.8867 | 0.7246 | 0.7474 | 0.7484 |
| 3 | 0.9224 | 0.8867 | 0.7248 | 0.7472 | 0.7479 |
| 5 | 0.9224 | 0.8866 | 0.7248 | 0.7476 | 0.7479 |

The effect of prepending metadata during training on metadata-probing accuracy and grammatical accuracy varies depending on the prompt length. It is natural to ask whether a similar phenomenon occurs in a pre-training task, that is, whether the performance of next-token prediction changes depending on the position within the sequence. Interestingly, such a phenomenon was not observed in next-token prediction. Indeed, as shown in Table B.1, the loss values calculated for each position also showed little variation across the depth of metadata, just as the overall loss remains almost unchanged.

## C  Calibration of the trained models

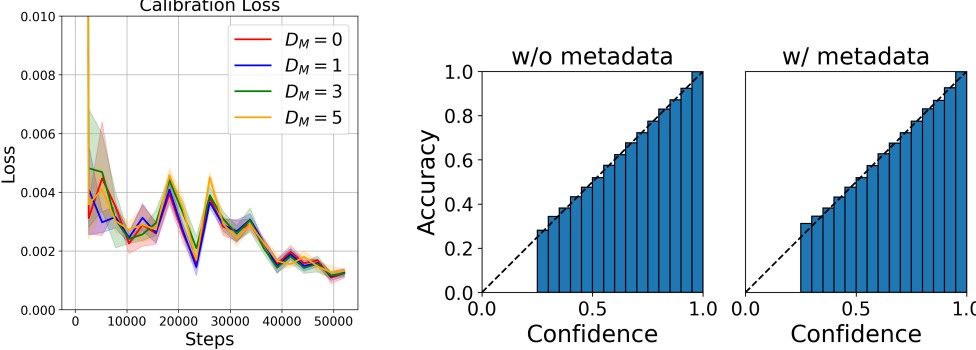

Figure C.1: Left: Change in calibration loss for test data (w/o metadata) with respect to the number of pre-training steps. Right: Calibration histogram at the final step of pre-training.

To reinforce the evidence that metadata conditioning does not affect the pre-training task when not prepending metadata, we investigate whether the models capture the distribution correctly. For this purpose, we introduce the metric called *expected calibration error*, which is defined in Minderer et al. (2021).

We provide the definition of the metric here. We first consider dividing the interval $[0, 1]$ into $m$ sub-intervals: $[0, 1/m), \ldots, [(m-2)/m, (m-1)/m), [(m-1)/m, 1]$. We then define $B_i$ $(i = 1, \ldots, m)$ as the set of all tokens in the test data for which the model's confidence falls within the $i$-th interval defined above. Expected Calibration Error (ECE) is defined as follows:

$$\text{ECE} = \sum_{i=1}^{m} \frac{|B_i|}{n} \cdot |\text{accuracy}(B_i) - \text{confidence}(B_i)| .$$

where $\text{accuracy}(B_i)$ and $\text{confidence}(B_i)$ denote the model's accuracy and the average confidence for these tokens, respectively. Intuitively, ECE measures the discrepancy between the model's accuracy and confidence. Therefore, a well-calibrated model that accurately predicts distributions is expected to have an ECE value close to zero.

We present the expected calibration error curve on the left side of Figure C.1. In both cases with and without metadata, we observe a decrease in expected calibration error, and there was no significant difference in calibration. The middle and right plots in Figure C.1 presents a histogram showing the average accuracy for each confidence interval. This also confirms that the model's confidence aligns with its accuracy, and the presence of metadata during training does not largely affect the model's calibration, as is the case with the test loss.

## D  More Details of Metadata-probing

**Training of Probing Classifiers.**   For training and evaluation of the classifier, we used 10,000 data points different from those used in pre-training. Specifically, 7,000 data points are used for training, 1,500 for validation of early stopping, and the remaining $n = 1,500$ for evaluation.

**Results for other layers.**   As we mentioned in Section 3.2, we conducted probing experiments on various layers. We omit the full results in the main paper, and present it in Figure D.1. We can observe that the results for the layers other than the 12th layer have similar tendencies to the result provided in Section 3.2.

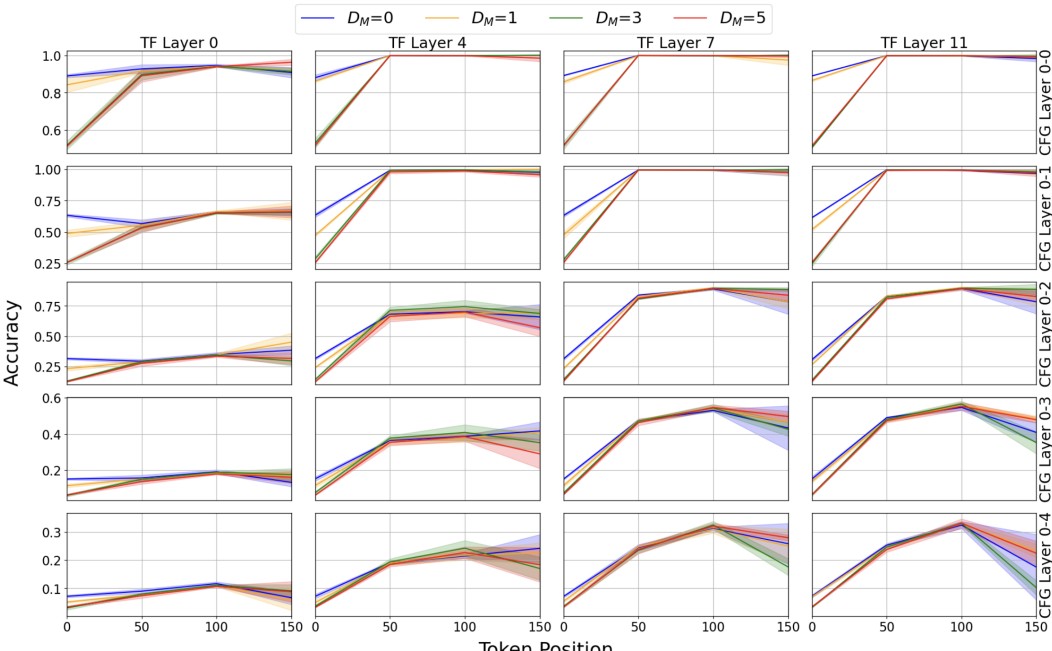

Figure D.1: Supplementary results of metadata-probing accuracy corresponding to Figure 3.2. We plot the transition with respect to the prompt length. Each column corresponds to a Transformer layer index $(1, 5, 8, 12)$, and each row corresponds to the depth of the CFG tree. The $d$-th row $(d = 1, 2, 3, 4, 5)$ represents the accuracy of classifying branches of the CFG tree from layer 0 up to layer $d - 1$. The chance rate of correctly predicting the metadata from layer 0 to layer $d - 1$ is $1/2^d$.

