# OpenReview forum: "When Does Metadata Conditioning (NOT) Work for Language Model Pre-Training? A Study with Context-Free Grammars"
_colmweb.org/COLM/2025/Conference — COLM 2025_

### Official Review · Reviewer_xHQg · 2025-05-12

**Rating:** 5
**Confidence:** 4
**Ethics Flag:** 1

**Summary:**

This paper investigates when providing metadata at the beginning of text documents during pre-training of large language models improves the results. The authors generate synthetic data using PCFGs for their experiments and conclude that prepending metadata is useful only when the context is long enough.

The goal of the paper is certainly interesting.Being able to understand when a technique will improve the results is a worthy goal. However, it is not clear to me whether the setup that the authors use is realistic. Using artificial data generated by PCFGs can be biased.

Furthermore, the paper lacks important details. For example, there is no information about the size of the data generated.

Finally, the section on the interpretation of the results and findings lacks concrete connections to the actual results (and contains no data).

**Questions To Authors:**

- Could you please clarify how much data is used? What is the effect of the amount of data?

- Is there any data to backup the interpretation provided in Section 4?

**Reasons To Accept:**

The main reason to accept this paper would be the attempt it makes to investigate the impact of prepending metadata to the training documents during pre-training.

**Reasons To Reject:**

The main reason to reject the paper is the limiting setup relying on PCFGs to generate artificial data. The overall setup is not realistic.

Furthermore, the paper lacks important details. For example, there is no information about the size of the data generated.

Finally, the section on the interpretation of the results and findings lacks concrete connections to the actual results (and contains no data).

---

> ### Author Response · Authors · 2025-06-03
> **Dear Reviewer xHQg,**
>
> Dear Reviewer xHQg,
>
>
> Thank you for your constructive feedback on our paper. We appreciate you recognizing the interesting goal of our work in investigating the impact of prepending metadata during language model pre-training.
>
> **Generalizability of PCFGs**: We appreciate your engagement with our rationale for using PCFGs to generate artificial data. For a detailed discussion, we kindly refer you to our **Official Comment by Authors** where we elaborate on this point. We believe this controlled approach provides valuable foundational insights into the underlying mechanisms.
>
> We also appreciate your comments regarding the need for specific details. Your feedback is helpful in improving the clarity of our presentation.
>
> - **Regarding data size**: We will ensure that the exact size of the datasets used in our experiments will be clearly stated in the revised version of the paper and our  **Official Comment by Authors**. Regarding the effect of data size, this study focuses on the role of metadata in learning, especially when the underlying structure is hard to capture without it. Thus, varying data size is beyond our scope. We believe our main conclusion—that learning PCFGs without metadata is difficult—would hold even with more data. In fact, [1] observed similar trends between a 100M-token (“finite setting”) and a 4.9B-token (“infinite setting”). Our 1B-token setup, already ten times larger than the former, suggests further scaling would not significantly change the outcome.
>
> - **Regarding Section 4**:  To clarify, Section 4 is not intended to present new hypotheses, but rather to offer interpretations of our empirical results in Section 3. We believe that our original manuscript did establish a connection from the data, figures, and tables to our explicitly stated "Finding 1" and "Finding 2", which then directly informed the interpretations presented in Section 4. However, we acknowledge that this link may not have been as explicit or clear as it could have been. To make our arguments clear, the table below outlines how the claims derived in §4 (our implications) correspond to the numerical outcomes (our results). Additionally, we will revise Section 4 to more directly and clearly connect our interpretations and results.
>
>
>
> | Claim derived in §4                                                                                           | Where it is tested                     | Numerical outcome                                                                                  |
> | :--------------------------------------------------------------- | :------------------------------------------------------ | :---------------------------------------------------------------------------- |
> | (P 1) Training with or without metadata should give *indistinguishable next-token loss* as long as inference is also done *without metadata*.                  | Figure 3.1, left pane                  | All four models (DM = 0, 1, 3, 5) converge to the same loss curve; differences are within noise.                                  |
> | (P 2) If the model is trained with metadata, its internal predictor $p(j\|x)$ stays weak, so *metadata-probing accuracy collapses on short prompts* and recovers only when the prompt becomes long. | Figure 3.2 & full-layer plots in App. D | For DM = 3,5 the probing accuracy at prompt length 5 is near chance, then rises steeply by length 50–100, exactly as predicted. |
> | (P 3) The downstream "grammatical-accuracy" task should suffer on short prompts but surpass the no-metadata baseline on long prompts.                       | Table 3.1                              | At prompt length 5, accuracy for DM = 3,5 plummets (0.072 and 0.000 vs 0.810). By length 25–50, the DM = 3 model overtakes the baseline. |
> | (Aux.) If metadata is present *at inference*, the loss should drop below the no-metadata level.                                                               | Figure 3.1, right pane                 | Loss with metadata heads well below the curve in the left pane.                                                                   |
> | (Aux.) No hidden positional effects in the language-model objective.                                                                                       | Table B.1                              | Loss at token positions 0–50 is flat across all DM settings.                                                                      |
>
>
>
> We value your feedback, as it helps us improve the clarity and completeness of our work. We hope these clarifications in our revision and official response will better demonstrate our study's contributions.
> We would be deeply appreciative if you would consider increasing the score once all issues are resolved.
>
> Thank you again for your review.
>
> Sincerely,
>
> Authors
>
> ----
>
> [1] Zeyuan Allen-Zhu, Yuanzhi Li. Physics of Language Models: Part 1, Learning Hierarchical Language Structures. arXiv Preprint arXiv:2305.13673. 2023.

---

> > ### Comment · Reviewer_xHQg · 2025-06-08
> >
> > After reading the author response and that other reviews, I have decided to increase my score.

---

### Official Review · Reviewer_StBC · 2025-05-13

**Rating:** 6
**Confidence:** 4
**Ethics Flag:** 1

**Summary:**

The paper investigates the effect of metadata conditioning in training language models. The authors conduct a series of experiments using probabilistic context-free grammars (PCFGs) to analyze how prepending metadata in training data influences the model's ability to learn semantic structures. They find that metadata conditioning improves performance in downstream tasks when task’s prompt is long, but degrades performance when the prompt is short and latent semantics cannot be inferred.

**Questions To Authors:**

- The metadata is provided starting from the top-level rules. Could the result be different if the metadata is provided starting from the bottom-level rules? The bottom-level rules are applied last in the generation process, so the model may be able to infer those rules easier from data.
- In line 223, it is not quite clear why "learning the richer posterior given metadata p(G_L|x, j) is facilitated". The posterior is sharper because of the metadata j, but it does not necessarily mean that the model is learning better.
- This also links to the conclusion in line 296 "When x is rich, ... it can implicitly construct p(G_L|x) more accurately". Even when x is long, predicting p(j|x) may still be imperfect, and this error will propagate to the posterior p(G_L|x, j). The advantage of modeling posterior through metadata j versus directly modeling thus seems not very clear.

**Reasons To Accept:**

- The paper provides an in-depth analysis of the effect of metadata conditioning, highlights the conditions under which it helps learning semantics and when it does not. This is a valuable contribution to understanding the role of metadata in training language models.
- The paper presents both clear empirical results with controlled interpretable data and a theoretical analysis, providing novel insight into the underlying mechanisms of metadata conditioning.

**Reasons To Reject:**

The experiments with PCFGs, although well-designed, still falls short in experimental rigor. Hyperparameters that could affect the results are not sufficiently explored, making it hard to draw strong conclusions from the results.
Some improvements to the experiments could strengthen the conclusions:
- Model size: due to the well-known phenomenon of "emergent abilities" in LLMs, larger models may exhibit different behaviors in learning semantics with or without metadata. The authors could consider testing models of different sizes.
- Large-scale pretraining: the authors should consider the impact of large-scale pretraining on the model's ability to learn semantics with metadata conditioning. Large-scale pretraining typically include a large amount of code data and structured data, which could significantly affect the model ability to model grammar in PCFG-generated data. For example, this can be done by performing continual pretraining using PCFG data on top of Meta's LLaMA model weights.
- Size of training data: the author mentioned that "Without metadata, the model needs to learn the hierarchical mixture of grammars directly from data, which is challenging due to the diverse structure of possible grammars" in Section 4. Data size is a critical factor determining whether the model can successfully learn the grammar without metadata. Therefore, the authors should consider training the model with different data sizes to see if the observation generalizes.
- Longer prompt length in Table 3.1: The results in Table 3.1 does not seem to show a consistent advantage of metadata conditioning even at longer prompt lengths. The authors could consider testing the model with longer prompt lengths, like in Figure 3.2, to check if the conclusion "for tasks with long prompts, metadata conditioning
improves the performance" (Finding 2) holds.

---

> ### Author Response · Authors · 2025-06-04
>
> Dear Reviewer StBC,
>
> We sincerely thank you for your thorough review and valuable feedback. We have carefully considered all your concerns and we appreciate the opportunity to address them.
>
> - **Regarding model size**: The effect of adding metadata on learning outcomes in larger model sizes is investigated in [1]. Specifically, it is reported that while adding metadata shows little impact on perplexity in a 1.6B model, performance on downstream tasks varies in models of 600M, 1.6B, 3B, and 8B sizes. We have confirmed that the phenomenon, where perplexity does not significantly change but downstream task performance does, is also reproduced in a smaller 100M-scale model. Accordingly, we focus on elucidating the mechanisms behind the phenomenon in this setting. Consequently, while we recognize that investigating potential differences in behavior resulting from varying model sizes is an interesting research direction, we consider it to be outside the direct scope of this paper.
>
> - **Regarding Large-scale pretraining**: The primary focus of this research is not to examine the effect of metadata in the process of continual learning. Instead, our aim is to verify the effect of metadata in standard pre-training (i.e., pre-training that does not assume any prior learning has been conducted). Therefore, we believe that the investigation into the relationship between continual learning and metadata, which you proposed, falls outside the direct scope of this research. However, we recognize your suggestion as an important consideration for future work.
>
> - **Regarding data size**: The primary objective of this research is to investigate the impact of the presence or absence of metadata on learning, particularly in situations where it is difficult to capture the full structure of data without metadata and we are not exploring the limits of learning capability without metadata. Therefore, we consider experiments with different data sizes to be outside the scope of this research. Furthermore, even if the training data size were increased, we believe it is unlikely that there would be a significant change to our main conclusion regarding the difficulty of grammar learning without metadata. [2] reports that similar trends were observed when comparing a 100M token dataset ("finite setting") with a 4.9B token dataset ("infinite setting") for PCFG learning. Our experiments used 1B tokens, which is 10 times the "finite setting," suggesting that further increases in data size are unlikely to significantly change learning outcomes.
>
> - **Regarding Table 3.1**: In Table 3.1, while there is a localized exception where performance with 5 metadata is not good at a prompt length of 50, a general trend is that metadata conditioning (metadata=1,3,5) performs worse than no metadata (metadata=0) at shorter prompt lengths, but then improves performance as prompt length increases. It is also crucial to consider our evaluation methodology: sequences from the evaluation data shorter than the given prompt length are excluded. As a result, increasing the prompt length means the evaluation focuses on longer, and therefore more challenging, data instances. Since increasing task difficulty can obscure metadata's benefits, we believe testing with longer prompts might not be conducive to clearly demonstrating its effect.
>
> - **Regarding how to assign metadata**: As you pointed out, bottom-level rules could potentially affect how easily the model infers rules from the data. However, since our primary focus was on verifying the effect of metadata's presence itself on learning, we currently do not have clear insights into the impact of different metadata assignments.
>
> - Regarding concerns about lines 223 and 296: Our framework in Section 4 provides a way to **interpret** the trade-offs observed in our experimental results (Observation, Finding 1, Finding 2 in Section 3), and is not intended to explain the underlying "why" of these learning mechanisms by itself. Therefore, if this framework appears unclear, we believe its clarity becomes evident when considered alongside the experimental results in Section 3. However, since we acknowledge that this connection might not be sufficiently clear, we will revise Section 4 to link the interpretations and results more directly. To help illustrate this connection, we will also post a table showing this correspondence in the following comment.
>
> Thank you once again for your insightful comments. We hope our responses have adequately addressed your concerns. We would be deeply appreciative if you would consider increasing the score once all issues are resolved.
>
> [1] Gao, Tianyu, et al. "Metadata Conditioning Accelerates Language Model Pre-training." arXiv preprint arXiv:2501.01956 (2025).
>
> [2] Allen-Zhu, Zeyuan, and Yuanzhi Li. "Physics of Language Models: Part 1, Learning Hierarchical Language Structures." arXiv preprint arXiv:2305.13673 (2023).

---

> > ### Author Response · Authors · 2025-06-04
> >
> > Dear Reviewer StBC,
> >
> > Further to our previous response, we are providing a table below to illustrate the correspondence between our experimental results in Section 3 and their interpretation within the framework presented in Section 4.
> >
> > | Claim derived in §4                                                                                           | Where it is tested                     | Numerical outcome                                                                                  |
> > | :--------------------------------------------------------------- | :------------------------------------------------------ | :---------------------------------------------------------------------------- |
> > | (P 1) Training with or without metadata should give *indistinguishable next-token loss* as long as inference is also done *without metadata*.                  | Figure 3.1, left pane                  | All four models (DM = 0, 1, 3, 5) converge to the same loss curve; differences are within noise.                                  |
> > | (P 2) If the model is trained with metadata, its internal predictor $p(j\|x)$ stays weak, so *metadata-probing accuracy collapses on short prompts* and recovers only when the prompt becomes long. | Figure 3.2 & full-layer plots in App. D | For DM = 3,5 the probing accuracy at prompt length 5 is near chance, then rises steeply by length 50–100, exactly as predicted. |
> > | (P 3) The downstream "grammatical-accuracy" task should suffer on short prompts but surpass the no-metadata baseline on long prompts.                       | Table 3.1                              | At prompt length 5, accuracy for DM = 3,5 plummets (0.072 and 0.000 vs 0.810). By length 25–50, the DM = 3 model overtakes the baseline. |
> > | (Aux.) If metadata is present *at inference*, the loss should drop below the no-metadata level.                                                               | Figure 3.1, right pane                 | Loss with metadata heads well below the curve in the left pane.                                                                   |
> > | (Aux.) No hidden positional effects in the language-model objective.                                                                                       | Table B.1                              | Loss at token positions 0–50 is flat across all DM settings.                                                                      |
> >
> >
> >
> > We hope this table helps clarify how our framework provides a way to understand the trade-offs and phenomena observed in our experiments.
> >
> > Thank you once again for your consideration.

---

> ### Comment · Reviewer_StBC · 2025-06-10
> **Post-rebuttal comment**
>
> Thank you for the explanation. I agree with some of the authors' arguments, although I still think providing more experiment results could significantly strengthen the conclusion of the paper.
> I prefer to keep my current score, as it is already on the positive side.

---

### Official Review · Reviewer_dKwt · 2025-05-17

**Rating:** 6
**Confidence:** 3
**Ethics Flag:** 2

**Summary:**

- The paper studies on "metadata preconditioning", a phenomenon where prepending metadata to pre-training sequences seem to improve language modeling, specifically downstream performance [1].
- The paper proposes to study this phenomenon via $D$-level probabilistic context-free grammar (D-PCFG), a framework proposed in earlier work [2] that can serve to generate controlled training sequences for language models.
  - Specifically, under D-PCFG the authors propose to use the selected production rules of the CFG as the "metadata" for the training sequence
- The paper then performs a series of experiments using the above controlled setup using D-PCFG, studying:
  1. how metadata preconditioning affects the test loss (finding: it doesn't affect it)
  2. whether metadata preconditioning helps models learn representations (hidden states) that contain information about the metadata, by using trained probes (finding: it doesn't help, and in fact makes probes worse)
  3. how metadata preconditioning affects downstream performance, as defined by the model's ability to generate grammatically correct sequences (finding: metadata preconditioning hurts if prompt is short, but helps if prompt is long)

Refs
- [1] https://arxiv.org/abs/2501.01956
- [2] https://arxiv.org/abs/2305.13673

**Questions To Authors:**

My main question would be whether there's anything else that the authors could add to make clearer the correspondance from the toy tasks to the real-world language modeling setting.

Minor:
  - L57: "As shown in (A1)" --> perhaps better explicitly refer to the bottom left plot

**Reasons To Accept:**

- A1. The paper studies a relevant problem, as a direct follow-up of Gao et al. (2025) [1] using the techniques from Allen-Zhu and Li (2023) [2].
- A2. The paper is well-written and well-executed. Experimental setup and takeaways are clear. The formatting of the paper is nice and clean.
- A3. The analysis provided in section 4 provides useful information of the conducted experiments.
- A4. The paper is strong along the "Understanding Depth, Principled Approach" dimension for reviewing guidelines (https://colmweb.org/ReviewGuide.html)

Refs:
- [1] https://arxiv.org/abs/2501.01956
- [2] https://arxiv.org/abs/2305.13673

**Reasons To Reject:**

R1. As with other attempts to reduce real-world natural language modeling into toy tasks, a general weakness of this work is that it is unclear why the result necessarily transfer to language modeling for natural text.
  - Yes, the PCFG + metadata construction is intuitive, but the exact correspondance to real-world data seems unclear
  - That is, for the claim in the conclusion ("metadata conditioning enhances performance on downstream tasks with sufficiently long task prompts that allow for the prediction of latent semantics, but it degrades performance on tasks with task prompts too short to predict latent semantics."), the paper justifies why this is the case for D-PCFG and makes an inductive reasoning step that this phenomenon must also extend to natural language. My concern is that this inductive step isn't valid.

R2. (As a result of R1) The paper would benefit from providing more examples and more intuition on the problem.
- While section 4 provides interpretations on the D-PCFG results, I'd be more interested in intuitions that map the provided setup to a realistic, natural language task.

R3. Even within the D-PCFG setup, it seems reductive to equate "Downstream Performance" in natural language modeling to "grammatical accuracy". I.e., the task setups feel contrived and detached from real-world settings.
- For example, one counter-argument is that Gao et al.  measures "downstream performance" on multiple tasks, whereas in this paper's setup there is only one possible downstream task

---

> ### Author Response · Authors · 2025-06-03
> **Dear Reviewer dKwt,**
>
> Dear Reviewer dKwt,
>
> Thank you for your detailed and constructive review of our paper. We appreciate your positive feedback on our paper's relevance, clarity, analysis, and principled approach.
>
> We also appreciate your thoughtful concerns regarding the generalizability of our D-PCFG findings to natural language (R1, R2) and our definition of "downstream performance" (R3).
>
> **Regarding Generalizability (R1 & R2)**:  We will provide a more detailed explanation regarding our approach to grammar complexity in our **Official Comment by Authors**, which we hope will further clarify our rationale.
>
> **Regarding Downstream Performance (R3)**: We also understand your point about equating "downstream performance" primarily with "grammatical accuracy" in our D-PCFG setup.
>
> Grammatical accuracy represents one of the most fundamental autoregressive generation capabilities, as investigated in [1]. Unlike next-token prediction loss, this metric allows us to assess phenomena such as the gradual degradation of sentence structure when generating long sequences. As grammatical correctness indicates an understanding of language structure, a model’s ability to produce well-formed and coherent sequences is foundational for any meaningful linguistic output.
>
> We believe that while natural language encompasses a broader range of downstream tasks, our chosen metric is a relevant and informative proxy for the specific aspects of metadata conditioning we aimed to study.
>
> **Minor Point**: Thank you for pointing out L57; we will revise "As shown in (A1)" to refer more explicitly to the specific plot for clarity.
>
> We are encouraged by your positive assessment. We hope that our clarifications and the further details in our official response will address your concerns. We would be sincerely grateful if you could consider raising the score once the concerns have been addressed.
>
> Thank you once again for your time and insightful feedback.
>
> Sincerely,
>
> Authors
>
> ----
>
> [1] Zeyuan Allen-Zhu, Yuanzhi Li. Physics of Language Models: Part 1, Learning Hierarchical Language Structures. arXiv Preprint arXiv:2305.13673. 2023.

---

> ### Comment · Reviewer_dKwt · 2025-06-06
> **Post-rebuttal comment**
>
> I thank the authors for the rebuttal. I have read the authors' responses and the other reviews. My current judgement is that:
> - I agree with both the strengths and weaknesses raised by the other reviewers
> - In particular, I share the concern with Reviewer xHQg arguing for rejection, that the paper's setup isn't very realistic.
> - Unlike Reviewer xHQg, however, I lean towards accepting the paper (for my listed reasons to accept) and believe that the paper may be of interest some parts of the community.
>
> I will maintain my current score.

---

### Official Review · Reviewer_d5Xy · 2025-05-19

**Rating:** 8
**Confidence:** 4
**Ethics Flag:** 1

**Summary:**

This paper investigates the impact of prepending metadata during language model pre-training using controlled experiments with artificial data generated from context-free grammars. The authors examined why previous research has shown inconsistent benefits from metadata conditioning, where improvements appear in some downstream tasks but not in the average next-token prediction loss.

Specifically, first, the researchers create synthetic data using probabilistic CFGs, where metadata represents information about which production rules were used to generate each sequence. Second, they train transformer models under different conditions, varying the metadata provided during pre-training. Finally, authors provided a theoretical framework explaining this phenomenon: models trained with metadata become better at predicting outputs given both context and metadata (p(y|x,j)) but worse at predicting metadata from limited context (p(j|x)). This creates the observed trade-off in performance depending on prompt length.

**Questions To Authors:**

See reasons to reject

**Reasons To Accept:**

- Well-designed controlled experiments: The use of synthetic data generated from CFGs allows for precise control over latent semantics and enables systematic investigation of how metadata conditioning affects model behavior.

- Evaluation: The authors use multiple metrics (next-token prediction, metadata-probing accuracy, grammatical accuracy) to evaluate model performance, providing a more complete picture of metadata's effects.

- "Theoretical" analysis framework: The paper develops a solid mathematical framework explaining why the observed effects occur, connecting empirical findings to theoretical understanding.

- Findings are interesting: The research provides actionable insights for practitioners, suggesting that the amount of metadata used during pre-training should be tailored to the expected context length in downstream tasks.

**Reasons To Reject:**

My major concern is the grammar complexity. The experiments use a relatively simple grammatical structure with binary choices at each level. While this simplification is justified for controlled experiments, it raises questions about generalizability to more complex real-world language structures. To that point, most of the data created or tasks are synthetic, hence I am not so sure how this method will generalize to real-world tasks.

---

> ### Author Response · Authors · 2025-06-03
> **Dear Reviewer d5Xy,**
>
> Thank you very much for your thoughtful review and positive assessment of our paper. We are grateful for your recognition of our controlled experiments, thorough evaluation, theoretical framework, and interesting findings.
>
> We appreciate your insightful comment regarding the grammatical complexity of our synthetic data and its generalizability to real-world language structures and tasks. We will provide a more detailed explanation regarding our approach to grammar complexity in our **Official Comment by Authors**, which we hope will further clarify our rationale.
>
> Thank you again for your valuable feedback and for recommending our paper for acceptance.
>
> Sincerely,
>
> Authors

---

> > ### Comment · Reviewer_d5Xy · 2025-06-09
> > **acknowledgement**
> >
> > thank authors for their response, I remain positive.

---

### Author Response · Authors · 2025-06-03

Dear Reviewers,

We would like to express our sincere gratitude for your thoughtful and constructive comments. In this general response, we address the concerns that were commonly raised by multiple reviewers.

**Justification of our synthetic dataset**: We believe that our D-level PCFG is a sufficient and appropriate setting for achieving our research objectives. First, it is well established that PCFGs can express a wide variety of languages (see, e.g., [1]). Additionally, previous studies have also utilized data generated using PCFG to investigate the pretraining of LLMs [2, 3, 4, 5, 6].

Furthermore, our experimental setting is designed to associate the metadata to various stuff observed in natural language. While one might intuitively think that, since we use synthetic data generated by a PCFG, the metadata only reflects grammatical differences, this is not actually the case.

For example, the metadata in our synthetic dataset can be considered analogous to the topic of a sentence or the URL from which a sentence originates. In natural language, changes in topic or source URL often lead to shifts in word frequency distributions. Similarly, in our setting, different metadata values correspond to different PCFG trees, resulting in changes in the frequency of terminal symbols.

We also posit that the metadata in our synthetic dataset can correspond to aspects such as the tone (e.g., explanatory vs. conversational) or genre (e.g., news, biography, dictionary/Wikipedia, or fiction) of natural language. In natural language, such changes often lead to grammatical shifts—e.g., more formal and complex structures in encyclopedic genres vs. simpler and more casual constructions in spoken or informal texts. Our PCFG-based setting replicates this: different metadata values lead to different underlying grammars, which are consistent within the data associated with a given metadata value, just as in natural language.

Thus, the metadata in our D-level PCFG setting captures a variety of metadata types conceivable in natural language. We therefore believe our dataset meaningfully reflects certain aspects of natural language metadata and is a valid framework for studying the effect of metadata conditioning. While it is synthetic, it is not an unrealistic abstraction.

Finally, we would like to emphasize that natural language is inherently complex, and capturing all of its properties is beyond the scope of our study. We intentionally used synthetic data with controllable metadata in order to isolate and investigate the effect of metadata in pre-training. Generalizing findings from controlled experiments to natural language is, in itself, an ambitious challenge and not the central aim of our work.

**Lack of description on the experimental settings**: We sincerely agree with the reviewers who pointed out that our paper lacks sufficient detail regarding the experimental settings. Below, we provide additional information, which we will incorporate into the revised version of the paper.
- The number of data samples used for pre-training is 500,000, and the training is conducted over a single epoch. We note that the difference in results between the models with and without metadata is not due to overfitting.
- For batch size, optimizer, and learning rate, we followed the setup in [2]. Specifically, we used a batch size of 96, AdamW optimizer with $\beta = (0.9, 0.98)$, weight decay of 0.1, and a learning rate of 0.0003.
- Details about the model size and D-level PCFG settings are provided in lines 128–137 of the paper.

We have also addressed reviewer-specific concerns individually in their respective responses. We would be happy to address any further questions or clarifications you may have.

Sincerely,

Authors

-----
[1] G. K. Pullum and G. Gazdar. Natural languages and context-free languages. *Linguistics and Philosophy 4* (1982).

[2] D. Hupkes, V. Dankers, M. Mul, E. Bruni. Compositionality decomposed: how do neural networks generalise? *Journal of Artificial Intelligence Research, 67, 757-795* (2020).

[3] S. Bhattamishra, K. Ahuja, N. Goyal. On the Ability and Limitations of Transformers to Recognize Formal Languages. In *Proceedings of the 2020 Conference on Empirical Methods in Natural Language Processing* (2020).

[4] Z. Allen-Zhu and Y. Li. Physics of language models: Part 1, learning hierarchical language structures. *arXiv preprint arXiv:2305.13673* (2023).

[5] H. Zhao, A. Panigrahi, R. Ge, S. Arora. Do Transformers Parse while Predicting the Masked Word? In *Proceedings of the 2023 Conference on Empirical Methods in Natural Language Processing* (2023).

[6] J. Jumelet, W. Zuidema. Transparency at the Source: Evaluating and Interpreting Language Models With Access to the True Distribution. In *Findings of the Association for Computational Linguistics: EMNLP 2023* (2023).

---

### Decision · Program_Chairs · 2025-07-08

**Decision:**

Accept

**Comment:**

This paper studies the impact of metadata conditioning in a synthetic dataset generated from PCFGs, where the latent variables are known and can be added as metadata. They vary and ablate the metadata and find some conclusions on the impact on test loss, metadata prediction, and impact on downstream performance in the context of prompt length. While reviewers felt that there could be more connections between the findings on PCFGs to real data, they noted that this was generally a clean and interesting setup to study this problem. Some reviewers felt that there could be a more comprehensive sweeps of different aspects such as data size and model size. Generally they felt that the paper provided interesting findings for the community, although only on synthetic setups.